# Polyethylenimine Grafted onto Nano-NiFe_2_O_4_@SiO_2_ for the Removal of CrO_4_^2−^, Ni^2+^, and Pb^2+^ Ions from Aqueous Solutions

**DOI:** 10.3390/molecules29010125

**Published:** 2023-12-24

**Authors:** Mehdi Khalaj, Seyed-Mola Khatami, Mehdi Kalhor, Maryam Zarandi, Eric Tobechukwu Anthony, Axel Klein

**Affiliations:** 1Department of Chemistry, Buinzahra Branch, Islamic Azad University, Buinzahra 14778-93855, Iran; 2Department of Chemical Industry, Technical and Vocational University (TVU), Tehran 14357-61137, Iran; 3Department of Chemistry, Payame Noor University, Tehran 19395-4697, Iran; 4Institute for Inorganic Chemistry, Department of Chemistry, Faculty of Mathematics and Natural Sciences, University of Cologne, Greinstrasse 6, 50939 Köln, Germany

**Keywords:** polyethyleneimine, core-shell NiFe_2_O_4_@SiO_2_, magnetic separation, toxic metals, adsorption

## Abstract

Polyethyleneimine (PEI) has been reported to have good potential for the adsorption of metal ions. In this work, PEI was covalently bound to NiFe_2_O_4_@SiO_2_ nanoparticles to form the new adsorbent NiFe_2_O_4_@SiO_2_–PEI. The material allowed for magnetic separation and was characterized via powder X-ray diffraction (PXRD), showing the pattern of the NiFe_2_O_4_ core and an amorphous shell. Field emission scanning electron microscopy (FE-SEM) showed irregular shaped particles with sizes ranging from 50 to 100 nm, and energy-dispersive X-ray spectroscopy (EDX) showed high C and N contents of 36 and 39%, respectively. This large amount of PEI in the materials was confirmed by thermogravimetry–differential thermal analysis (TGA-DTA), showing a mass loss of about 80%. Fourier-transform IR spectroscopy (FT-IR) showed characteristic resonances of PEI dominating the spectrum. The adsorption of CrO_4_^2−^, Ni^2+^, and Pb^2+^ ions from aqueous solutions was studied at different pH, temperatures, metal ion concentrations, and adsorbent dosages. The maximum adsorption capacities of 149.3, 156.7, and 161.3 mg/g were obtained for CrO_4_^2−^, Ni^2+^, and Pb^2+^, respectively, under optimum conditions using 0.075 g of the adsorbent material at a 250 mg/L ion concentration, pH = 6.5, and room temperature.

## 1. Introduction

Toxic metals such as lead have been used by humans for thousands of years, but only with the industrial revolution and the rapid growth of the human population and industrial activities in the last 70 years has the penetration of toxic metals into the natural environment and water resources enormously increased, and this represents a threat to human health [1]. Many of the metals of concern are heavy metals such as Hg, Cd, and Pb. This is probably why the term “heavy metals” is frequently but wrongly used for all toxic metals. However, toxic metals such as Be, Cr, and Ni should not be termed heavy metals as their density does not exceed 5 g/cm^2^ and their chemistry is also dissimilar to heavy metals that have a high binding affinity to sulfur-based (bio)ligands in common [1,2].

The detection of toxic metals and their removal from water resources has continuously been an important scientific topic, and various methods have been developed to remove toxic metals, such as chemical precipitation (including coagulation and flocculation), adsorption, electrochemical reduction, removal through membrane processes, reverse osmosis, and ion exchange methods [3]. Important goals are reducing costs, simplifying methods, avoiding double contamination, and increasing sensitivity [4,5,6,7]. Amongst these methods, adsorption (chemi- and physisorption) is the most interesting in terms of sensitivity and selectivity, and functionalized nanoparticular materials seem to be favorable due to their large surface-to-mass ratio [4,6,8,9,10,11,12,13,14].

From a chemical viewpoint, the main challenges in the removal of toxic metal ions from wastewater lie in the efficiency and recyclability of adsorbents [3]. Polyethyleneimine (PEI) has been reported to be an efficient material that shows fast uptake and fast release under different pH conditions [15,16,17,18,19]. For recovery, most adsorbents were separated and then recycled through centrifugation or filtration, but in recent years, the use of magnetically separable adsorbent materials has been introduced and seems very promising in order to achieve high recycling rates [5,17,20,21,22,23]. Frequently, the magnetic separation process is based on hematite and magnetite embedded in core-shell nanoparticles [17,20,24,25,26,27], and in recent years, hematite and magnetite structures have been successfully replaced with ferrite structures such as nickel ferrite (NiFe_2_O_4_) [28,29], CoFe_2_O_4_ [25,30], or MnFe_2_O_4_ [21,31].

Magnetically separable core-shell Fe_3_O_4_@SiO_2_ nanoparticles, functionalized with PEI and 1,4,5,8-naphthalenetetracarboxylic-dianhydride (NTDA), were recently used to adsorb Pb^2+^ ions in the presence of Cd^2+^, Ni^2+^, Cu^2+^, and Zn^2+^ [17]. Further similar materials were Fe_3_O_4_@MIL-88A(Fe)–APTMS NP based on the Fe-containing metal-organic framework MOF MIL-88A(Fe) and (3-aminopropyl)trimethoxysilan (APTMS) applied for the removal of CrO_4_^2−^, Cd^2+^, and Pb^2+^ [32], CoFe_2_O_4_@MWCNT–CTS NP based on multi-walled carbon nanotubes and chitosan (CTS) for the adsorption of Pb^2+^ [30], MnFe_2_O_4_@GO–TPA based on graphene oxide (GO) and tetraethylenepentamine (TPA) for the adsorption of Pb^2+^ [31], and very recently Fe_3_O_4_@SiO_2_–CTS–DTPA with diethylenetriaminepentaacetate (DTPA) binding at the NH functions of CTS for the removal of Pb^2+^ [33]. A similar comparative study for CrO_4_^2−^, Ni^2+^, and Pb^2+^ (along with Cd^2+^ and Hg^2+^) was previously conducted using amino-functionalized Fe_3_O_4_@GS nanomaterials based on non-further defined graphene (GS) [34]. CrO_4_^2–^ was efficiently removed using sodium lignosulfonate/PEI/sodium alginate beads very recently [35]. Very recently, polyaniline-grafted pine sawdust was used to efficiently adsorb Cu^2+^, Co^2+^, Cd^2+^, Ni^2+^, Pb^2+^, Zn^2+^, and Fe^2+^ in a comparative study [36]. In a very recent approach, 8-chloroacetyl–aminoquinoline (CAAQ) was attached through PEI as a ligand to Fe_3_O_4_@SiO_2_ nanoparticles for the capture of Fe^3+^, Cu^2+^, and Cr^3+^ [23].

Herein, we report a study on the use of polyethylene imine (PEI) grafted onto core-shell NiFe_2_O_4_@SiO_2_ nanoparticles (Figure 1) as an adsorbent for the removal of CrO_4_^2−^, Ni^2+^, and Pb^2+^ ions from water. We studied the influence of parameters such as pH, temperature, metal ion concentration, and the amount of a NiFe_2_O_4_@SiO_2_–PEI adsorbent and also investigated the kinetics of the adsorption system.

## 2. Results and Discussion

### 2.1. Characterization of the Adsorbent

The powder XRD pattern of the NiFe_2_O_4_@SiO_2_–PEI adsorbent showed signals at 2Ɵ (assigned *hkl* values) = 30.0 (*220*), 37 (*311* + *222*), 43.4 (*400*), 53.7 (*422*), 57.7 (*511*), 62.9 (*440*), 71.4 (*620*), and 74.6° (*533*) characteristic of the cubic phase of nickel ferrite (NiFe_2_O_4_, reference code: 00-003-0875) (Figure 1). A further signal at 46.8° could not be assigned. Reflections corresponding to crystalline silica were absent, but we assigned the broad features from 12 to 42° and from 50 to 78° to the amorphous SiO_2_–PEI shell, in keeping with a relatively large total of about 1 g SiO_2_ + PEI on 2 g of NiFe_2_O_4_ (see synthesis). In the recovered NiFe_2_O_4_@SiO_2_–PEI adsorbent, all PXRD features were retained.

In contrast with PXRD, the FT-IR analysis of NiFe_2_O_4_@SiO_2_–PEI (Figure 2) also revealed the amorphous part, with bands at 3678, 3609, and 3573 cm^−1^ assigned to N–H stretching vibrations and the broad peak between 3100 and 3700 cm^−1^ assigned to O–H stretching modes, while C(sp^3^)–H stretches appear sharp at 2945 and 2879 cm^−1^. The C–H bending modes are found at 1348 cm^−1^. The bands located at 1634, 1210, 1153, 1110, 1038, 924, and 879, cm^−1^ can be assigned to the Si–O–Si, Si–O, C–C, C–N, and C–O functionalities [28,35]. Finally, the NiFe_2_O_4_ core of the material causes the Ni/Fe–O lattice vibrations to appear as a broad band centered at 530 cm^−1^ [23,28,35].

The field emission scanning electron microscopy (FE-SEM) images of NiFe_2_O_4_@SiO_2_–PEI showed irregularly shaped and partially agglomerated particles with diameters ranging from 50 to 100 nm (Figure 3), similar to what we recently reported for NiFe_2_O_4_@SiO_2_–PSA particles (PSA = propylsulfonic acid) that were prepared in a similar manner [28]. The contrast of all particles is identical which is in line with the complete surface of the initial NiFe_2_O_4_ particles covered with large amounts of SiO_2_ and PEI, in keeping with the PXRD results. Energy-dispersive X-ray spectroscopy (EDX) analysis showed C (36.31%), N (38.96%), Ni (2.12%), O (6.78%), Fe (4.38%), and Si (4.12%) (Figure 3) and thus confirms the relatively large amounts of SiO_2_ compared to Fe and Ni, which were found in the correct 2:1 ratio. Based on PEI (~CH_2_CH_2_NH), the weight ratio of C:N should be 24:14. However, the EDX analysis shows a C:N = 36:39. We suspect that the EDX method overestimates the surface of the material, which is dominated by the end-NH_2_ groups, thus producing the high values for N compared with C. The overestimation of the surface by EDX is supported by the overall large amounts of C and N originating from surface-bound PEI compared with the core-shell elements Ni, Fe, and Si.

Thermogravimetric (TGA) and differential thermal analysis (DTA) showed a small weight loss of about 5% around 100 °C. As we used a sample dried at 50 °C in vacuo, we assigned the corresponding endothermic peak to a loss of residual water bound in the PEI. The major weight of about 77% of the original mass occurred in the range of 200 to 500 °C (Figure 4). The residual 18% represents the NiFe_2_O_4_@SiO_2_ nanoparticles without the “organic” functionalization, which is in excellent agreement with the EDX analysis showing a total of 75% for C and N and thus a large coverage of the particles with PEI.

### 2.2. Adsorption Studies for CrO_4_^2−^, Ni^2+^, and Pb^2+^ Ions

#### 2.2.1. Effect of pH

In order to evaluate the effect of pH on the adsorption of CrO_4_^2−^, Ni^2+^, and Pb^2+^, the pH was varied from 3 to 8, while the other parameters were fixed at a 250 mg/L metal ion initial concentration, 0.075 g of adsorbent, 50 mL volume, and 298 K. The adsorption capacity showed a maximum at a pH of 6.5 and decreased with increasing pH (Figure 5). In acidic media at pH < 6, we assume competition between protons (H^+^) and the metal ions Ni^2+^ and Pb^2+^ in their coordination with the NH_2_ groups of the adsorbent. The maximum was reached at pH = 6.5 with adsorption capacities of 149.3, 156.7, and 161.3 mg/g for CrO_4_^2−^, Ni^2+^, and Pb^2+^, respectively.

Very similar pH-dependent behavior as our materials with the maximum adsorption at pH = 6 was previously reported for the Pb^2+^-adsorbing materials Fe_3_O_4_@SiO_2_@PEI–NTDA [17], PEI-bacterial cellulose [19], and sodium alginate (ALG)/PEI composite hydrogels [16], in line with PEI acting as a coordinating agent in these materials. However, also for Fe_3_O_4_@SiO_2_@PEI–CAAQ (CAAQ = 8-chloroacetyl–aminoquinoline) in which CAAQ acts as an additional ligand [23], the same behavior was found. The previously reported adsorbent Fe_3_O_4_@SiO_2_–CTS–DTPA (DTPA = diethylenetriaminepentaacetate) [33] already shows maximum Pb^2+^ adsorption in acidic solutions at pH = 3, no loss of binding capacity between pH = 3 and pH = 6, and an adsorption capacity of around 105 mg/g at pH = 6, which is markedly lower compared with our adsorbent material.

On the other hand, the very similar behavior of our adsorbent towards the cations Ni^2+^ and Pb^2+^ on one side and the anionic CrO_4_^2−^ is peculiar compared to other amine-containing materials such as the previously reported sodium lignosulfonate/PEI/sodium alginate beads [35], the ethylenediamine-functionalized Fe_3_O_4_ (EDA@Fe_3_O_4_) particles [37], amino-functionalized Fe_3_O_4_@GS nanomaterials [34], polydopamine modified chitosan aerogels [38], or the MOF APTMS@MIL-88A(Fe) (APTMS = (3-aminopropyl)trimethoxysilan) [32]; better CrO_4_^2–^ adsorption was found at low pH (2 to 3) while cation adsorption is superior at higher pH. This is reasonable in view of the protonated amine functions at low pH allowing to strongly adsorb the CrO_4_^2–^ anion while the neutral amine function coordinates cations. The only explanation we have so far is that the CrO_4_^2−^ was largely reduced to Cr^3+^ ions which would then absorb in a similar way to Ni^2+^ and Pb^2+^. This idea is supported by several reports that show that Cr^3+^ can be formed from CrO_4_^2–^ through electron transfer from various materials [15,18,26,35,38,39,40,41]. Such CrO_4_^2–^ to Cr^3+^ reduction upon adsorption can be very efficient if a distinct electron-donating material is present as in the CTAB-intercalated MoS_2_ nanosheets (CTAB = cetyl trimethyl ammonium bromide) that can be used for the simultaneous removal of Cr(IV) and Ni(II) [42] or in the chitosan-modified multi-walled carbon nanotube composites (MWCNT-CTS) that adsorb CrO_4_^2–^ exclusively as Cr^3+^ [43]. In future studies, we will use X-ray photoelectron spectroscopy (XPS) to study the oxidation states of the adsorbed Cr as was carried out in the last two mentioned studies.

The approximately 145 mg/g total adsorption capacity of our adsorbent for Cr compares to about 290 mg/g for the Fe_3_O_4_@SiO_2_@PEI–CAAQ adsorbent [32] which is only outnumbered by the 340 mg/g reported for an EDTA-inspired polydentate hydrogel [44]. In view of the additional mass of the metal oxide cores of the Fe_3_O_4_@SiO_2_@PEI–CAAQ and our adsorbent and the easy magnetic separation, the core-shell systems are superior even in capacity.

The capacity for Ni^2+^ absorption of about 150 mg/g found for our adsorbent material compares well with the amino-functionalized Fe_3_O_4_@GS nanomaterials [34].

#### 2.2.2. Effect of Contact Time

Examining the effect of contact time in the adsorption process of the metals allowed us to calculate the reaction rate and the time to reach equilibrium. For this purpose, the reaction parameters were kept constant with an initial concentration of metal salts of 250 mg/L, 0.075 g of adsorbent, and pH = 6.5. The adsorption capacity increased rapidly within the first 5 min, at a high rate within the first 20 min, and then continued slowly. Equilibrium was reached within 45 min (Figure 6).

For the previously reported similar materials Fe_3_O_4_@SiO_2_@PEI–NTDA [17] and Fe_3_O_4_@SiO_2_@PEI–CAAQ [23], the adsorption capacities reached plateau values only after more than 200 min [17] or 90 min [23], respectively, which indicates that our system is markedly more active and lies in the same time range as the previously reported PEI-bacterial cellulose [19]. In contrast to this, very fast adsorption of Cu^2+^, Co^2+^, Cd^2+^, Ni^2+^, Pb^2+^, Zn^2+,^ and Fe^3+^ within 10 to 20 min was achieved with polyaniline grafted onto pine sawdust [36], underlining the suitability of polyamines and anilines in efficiently coordinating the metals.

### 2.3. Adsorption Kinetics and Mechanism

The mechanism of the adsorption of the metal ions was studied via different kinetic models, e.g., pseudo-first order, pseudo-second order, and Elovich models [45]. The correlation coefficient (*R*^2^) values for the different kinetic models were calculated by drawing log(q_e_ − q_t_) vs. *t* (pseudo-first order), *t*/q_t_ vs. *t* (pseudo-second order), and q_t_ vs. ln *t* (Elovich) diagrams (Table 1). The agreement with pseudo-first-order kinetics is slightly better than the pseudo-second-order fit and much better than with the Elovich equation. This stands in contrast to the behavior of the reported adsorption of Pb^2+^ by an activated carbon [45] and we ascribe this to the more unspecific surface of the carbon in contrast to the well-defined coordination sites of PEI. This is supported by the very similar behavior of the Fe_3_O_4_@SiO_2_@PEI–NTDA [17] which also showed pseudo-first-order kinetics for the Pb^2+^ adsorption. The better agreement of experimental data with pseudo-second-order kinetics reported for Fe_3_O_4_@SiO_2_@PEI–CAAQ [23] is in line with the additional CAAQ ligand showing superior binding to PEI.

#### 2.3.1. Effect of the Amount of Adsorbent

The effect of the amount of adsorbent was studied in the range from 0.01 g to 0.1 g while the other parameters were held constant (conc. of adsorbates: 250 mg/L and pH = 6.5). With increasing amounts of adsorbent, the adsorption capacity increased up to about 0.08 g (Figure 7). The further increase did not give higher adsorption. The slight decrease in adsorption capacity at high adsorbent loads might be due to the aggregation and accumulation of particles and the overall reduction in their surface.

A marked maximum adsorption maximum for Pb^2+^ was found in the dosage behavior for the recently reported adsorbent material Fe_3_O_4_@SiO_2_@PEI-NTDA [17]. For this material as well, aggregation was assumed to be responsible for this phenomenon. When comparing the two curves, our maximum is less pronounced, meaning that our system is more tolerant of larger amounts of adsorbent.

#### 2.3.2. Effect of the Metal Ion Concentration

In order to determine the maximum adsorption capacity of the adsorbent, the effect of different concentrations of metal ions was evaluated. Figure 8 shows that the adsorption capacity of the adsorbent increases with the increase in the initial concentration of CrO_4_^2−^, Ni^2+^, and Pb^2+^ ions. The highest adsorption capacity was observed at a concentration of 250 mg/L. At higher concentrations, the adsorption capacity of the adsorbent remains constant, pointing to the saturation of the adsorbent sites.

### 2.4. Adsorption Isotherms

Equilibrium isotherm studies can provide information about the nature of the interaction between the adsorbed material and the adsorbent and can be used to determine the adsorption capacity of the adsorbent. In order to produce a view of the path of CrO_4_^2−^, Ni^2+^, and Pb^2+^ ions’ adsorption, the mechanism was investigated by applying the linear forms of Langmuir, Freundlich, and Temkin isotherm models [46]. The calculated parameters for different isotherms are depicted in Table 2. By looking at the parameters of the isotherms we are able to gain an insight into the adsorption mechanism. The Langmuir isotherm model is based on the hypothesis that a single layer of adsorbent material on the surface structure of the adsorbent is saturated during adsorption, the adsorption sites are identical, the energy of the adsorption is not dependent on the surface coverage, and there is no interaction between the adsorbates (here, the adsorbed metal ions) [45].

The correlation coefficient (*R*^2^) was calculated for all isotherms and fitted to the experimental data. Values of 0.972 (CrO_4_^2−^), 0.976 (Ni^2+^), and 0.997 (Pb^2+^) *R*^2^ show that the Langmuir isotherm fitting agrees very well with the experimental results. Accordingly, the mechanism of adsorption is monolayer adsorption on the surface of the adsorbent [46,47]. The same behavior was also found for similar adsorbent materials such as Fe_3_O_4_@SiO_2_@PEI–CAAQ [23], Fe_3_O_4_@SiO_2_@PEI–NTDA [17], and Fe_3_O_4_@SiO_2_–CTS/DTPA [33], while for the CrO_4_^2–^ adsorption on sodium lignosulfonate/PEI/sodium alginate beads [35], the Langmuir and Freundlich models gave very similar *R*^2^ values. The Freundlich isotherm applies to non-ideal adsorption on heterogeneous surfaces [48,49] and the Freundlich-type behavior is in line with the very heterogeneous surface of the lignosulfonate/PEI/sodium alginate material [35] in contrast with our adsorbent.

### 2.5. Adsorption Thermodynamics

The effect of temperature on the adsorption capacity was investigated to determine the thermodynamic parameters and investigate the spontaneity of the adsorption process. The adsorption capacity decreased with increasing temperature from room temperature to 75 °C (Figure 9). This is in line with the exothermic nature of the adsorption process.

From these data, we also calculated the Gibbs free energy (ΔG), enthalpy (ΔH), and entropy (ΔS) of the system. The change in ΔG of the CrO_4_^2−^, Ni^2+^, and Pb^2+^ ions at six different temperatures was determined through the relationships between ΔG, ΔH, ΔS, and ln Kc in the equations shown in Table 3. ΔH and ΔS were obtained by plotting ln Kc against 1/*T* (Figure 10).

The thermodynamic parameters ΔG, ΔH, and ΔS are shown in Table 3. The negative ΔG values are in line with spontaneous adsorption processes on the surface of the adsorbent. The negative values for ΔH show the exothermic nature of the adsorption reaction, which is very probably binding to the amine functions of the PEI. The negative ΔS values point to a non-spontaneous reaction. However, the overall exothermic binding is in line with rapid binding under these conditions (compared in Figure 6) and is an important pre-requisite for the use of this material for efficient metal recovery from solution.

As in the pH-dependent experiments, the anionic CrO_4_^2−^ behaves remarkably similar to the Ni^2+^ and Pb^2+^ cations with negative ΔH^0^, ΔS^0^, and ΔG^0^. This stands in contrast to the related sodium lignosulfonate/PEI/sodium alginate beads for which the CrO_4_^2−^ adsorption showed positive ΔH^0^ (7.5 kJ/mol) and ΔS^0^ (70 J/K·mol) values but a negative ΔG^0^ of −13.36 kJmol^−1^ at 298 K [35]. This difference supports our assumption that the CrO_4_^2−^ ions in our test solutions are adsorbed as Cr^3+^ ions on the adsorbent.

### 2.6. Scanning Electron Microscope (SEM)/Energy-Dispersive X-Ray (EDX) Analysis

Figure 11A shows the SEM image of the as-prepared NiFe_2_O_4_@SiO_2_–PEI adsorbent as nanoparticles of approximately 50 to 100 nm but more agglomerated than in Figure 3. The morphology does not change upon loading with Cr(VI), Pb(II), and Ni(II) (Figure 11B–D). The EDX spectra show the characteristic peaks of Cr(VI), Pb(II), and Ni(II) ions (Figure 11F–G). For comparison, we recorded the EDX of a recovered NiFe_2_O_4_@SiO_2_–PEI sample, and we found traces of Na^+^ and Cl^–^ (Figure 11E) stemming from the washing procedure (first HCl and then NaOH; see Materials and Methods, Section 3).

### 2.7. Adsorbent Recovery

The recyclability was tested in 11 consecutive runs, and the adsorbent showed good recovery (Figure 12) for all three ions.

For the previously reported Fe_3_O_4_@SiO_2_@PEI–CAAQ (CAAQ = 8-chloroacetyl–aminoquinoline) [23], efficient recycling was only achieved when using Na_2_EDTA^2+^ solutions for the stripping of the metal cations, while desorption using HCl or HNO_3_ steadily decreased the adsorption capacity. This underlines that the additional CAAQ ligand helps to more strongly bind metal cations but at the same time is detrimental to rapid and efficient desorption.

## 3. Materials and Methods

### 3.1. Instrumentation

Powder X-ray diffraction (PXRD) measurements were carried out on a Shimadzu 6100 using Cu-Kα (λ = 0.15406 Å) radiation at 298 K on solid powder samples of freshly prepared and recovered materials of NiFe_2_O_4_@SiO_2_–PEI. The PXRD of NiFe_2_O_4_ was recorded on an STOE-STADI MP diffractometer equipped with a Cu-Kα_1_ radiation (λ = 0.15406 Å) source and operating in transmission mode. Thermogravimetry–differential thermal analysis (TGA-DTA) was recorded on a Shimadzu TGA-DTG-60H instrument on a powder sample. Fourier-transformed (FT)-IR spectra were recorded on a Bruker Alpha I spectrophotometer on KBr disks. Field emission scanning electron microscopy (FE-SEM) and energy-dispersive X-ray spectroscopy (EDX) were carried out on a JEOL JSM-IT 100 instrument.

### 3.2. Reagents

All chemicals including NiCl_2_·6H_2_O (Merck, Darmstadt, Germany, puriss.p.a. >98%), FeCl_3_·6H_2_O (Merck, reagent grade > 98%), FeCl_2_·6H_2_O (Merck, puriss.p.a. > 99%), tetraethylorthosilicate Si(OEt)_4_ (TEOS) (Merck, reagent grade 98%), trimethoxy(3-(oxiran-2-ylmethoxy)propyl)silane (Sigma-Aldrich, St. Louis, MO, USA, M_W_ = 248.35 g/mol, 99%), PEI (Sigma-Aldrich, 5000 average molecular weight, 99%), and toluene (Merck, anhydrous 99.8%) were used without further purification.

### 3.3. Synthesis of the NiFe_2_O_4_@SiO_2_–PEI Adsorbent

First, NiFe_2_O_4_@SiO_2_ NPs were prepared according to our previously reported work [28]. In brief, NiFe_2_O_4_ NPs were produced by adding a mixture of 160 mL 1 M aqueous FeCl_3_·6H_2_O and 40 mL 1 M of NiCl_2_·6H_2_O quickly to 1 L of a boiling aqueous solution of 1 M NaOH under vigorous stirring. Then, the solution was cooled to room temperature and stirred continuously for 90 min. The resulting precipitate was then purified during four repeated washing–centrifugation–decantation cycles, each using 50 mL of water. 2 g of the NiFe_2_O_4_ NPs were dispersed in 25 mL EtOH by ultrasonic treatment for 2 h at 60 °C, and then, 10 mL 25% aqueous ammonia was added to the mixture and stirred at 60 °C for 40 min. Then, 1 mL of TEOS was added, and stirring was continued at the same *T* for another 24 h. The suspended silica-coated particles were separated from the solution by placing an external magnet in the flask and decanting the supernatant solution. The NPs were washed 3x with 15 mL MeOH and dried in vacuum for 48 h. Finally, the NiFe_2_O_4_@SiO_2_ NPs were calcinated at 800 °C for 2 h. 

For the preparation of the NiFe_2_O_4_@SiO_2_–PEI adsorbent, 10 g of PEI was dissolved in 50 mL of hot toluene and then cooled. This was mixed with 472 mg (2 mmol) trimethoxy(3-(oxiran-2-yl-methoxy)propyl)silane, and the mixture was heated under reflux for 8 h. Then, 2.5 g of the NiFe_2_O_4_@SiO_2_ nanoparticles was added at room temperature, and the mixture was heated under reflux for 5 h. The resulting colorless solid was filtered, washed with toluene, and dried at 50 °C in vacuo affording 2.9 g of NiFe_2_O_4_@SiO_2_–PEI adsorbent.

### 3.4. Adsorption Experiments

For the metal standard solutions, 5, 10, 15, 20, 25, and 30 mg/L of Na_2_Cr_2_O_7_·2H_2_O (for CrO_4_^2−^, M_W_ = 297.99 g/mol), Ni(NO_3_)_2_·6H_2_O (for Ni^2+^, M_W_ = 229.54 g/mol), and Pb(NO_3_)_2_·6H_2_O (for Pb^2+^, M_W_ = 370.84 g/mol) were dissolved in 100 mL deionized water. This translates to 0.3356–2.0135 mmol/L for CrO_4_^2−^, 0.2178–1.3070 mmol/L for Ni^2+^, and 0.1348–0.8090 mmol/L for Pb^2+^. The pH was adjusted to 3 to 8 using diluted NaOH (1 M) or HCl (1 M) solutions. Adsorption procedure: a 250 mL Erlenmeyer flask was supplied with 50 mL of each metal ion solution (50–300 mg/L), the adsorbent (0.01–0.1 g) was added, and the mixture was stirred at room temperature for 45 min at pH values ranging from 3 to 8.

The adsorption capacity at equilibrium (q_e_) and the adsorption capacity at time *t* (q_t_) are defined as:q_e_ = ((*C*_0_ − *C*_e_)*V*)/m           q_t_ = ((*C*_0_ − *C*_t_)*V*)/m
where *C*_0_: initial concentration (mg/L), *C*_e_: equilibrium concentration (mg/L), *C*_t_: concentration at the time *t*, m: amount of adsorbent (g), and *V*: volume of the solution (L) [50].

### 3.5. Adsorbent Recovery

After the adsorption of the CrO_4_^2−^, Ni^2+^, and Pb^2+^ ions at pH = 6.5, using 0.075 g adsorbent and 250 mg/L metal ions at 298 K for 45 min, the magnetic adsorbent was separated from the reaction batch with the help of an external magnet. For the recovery of the adsorbent material, the adsorbed metals were removed through means of washing with HCl solution (5%) followed by NaOH solution (5%).

## 4. Conclusions

A new adsorbent material NiFe_2_O_4_@SiO_2_–PEI, which is polyethylene imine (PEI) grafted on core-shell NiFe_2_O_4_@SiO_2_ nanoparticles, was synthesized through a simple and easy procedure and characterized using PXRD, FE-SEM, EDX, FT-IR, and TGA-DTA analyses. The potential of NiFe_2_O_4_@SiO_2_–PEI in the adsorption of CrO_4_^2−^, Ni^2+^, and Pb^2+^ ions from aqueous solutions was investigated under variation of pH, adsorbent amount, metal ion concentration, and temperature. The maximum adsorption was achieved at pH = 6.5 and a 250 mg/L CrO_4_^2−^, Ni^2+^, and Pb^2+^ ion concentration and 0.075 g of adsorbent at room temperature. The adsorption mechanism was investigated using pseudo-first-order, pseudo-second-order, and Elovich models with the best match of the pseudo-first-order model with the experimental results. The best fit for the adsorption isotherms was the Langmuir model, and both findings are in line with smooth homogeneous mono-layer adsorption. The adsorption of both the anionic CrO_4_^2−^ and the cation Ni^2+^ and Pb^2+^ ions increased with increasing time and decreased with increasing temperature. Deconvolution of the *T*-dependent adsorption gave negative values for ΔG, ΔH, and ΔS. For the metal cations Pb^2+^ and Ni^2+^, this is in line with the binding of these cations to the amine functions of the PEI. The very similar behavior of the anionic CrO_4_^2−^ is probably due to the reduction of CrO_4_^2−^ to Cr^3+^ which shows comparable binding properties to Pb^2+^ and Ni^2+^. In future studies, we will elaborate on this using XPS for the determination of the oxidation states of the Cr species bound to the adsorbent.

For the moment, we can state that the new adsorbent material NiFe_2_O_4_@SiO_2_–PEI is an interesting candidate for the removal of toxic metals from wastewater, in view of its simple preparation, simple adsorbing kinetics, exothermic thermodynamics (chemical binding), and good recovery and recyclability. In the future, we will further explore its potential by studying the adsorption of further metal cations such as Cu^2+^, Cr^3+^, and Gd^3+^ as well as the co-dependence of the adsorption of toxic metals with other cationic and anionic components in wastewater.

## Data Availability

The data presented in this study are available on request from the corresponding author.

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
