# Peer review of "Polyethylenimine Grafted onto Nano-NiFe2O4@SiO2 for the Removal of CrO42−, Ni2+, and Pb2+ Ions from Aqueous Solutions"

_molecules, 2023, doi:10.3390/molecules29010125_

Round 1

Reviewer 1 Report

Comments and Suggestions for Authors

In this paper, the preparation of polyethyleneimine covalently bound to NiFe2O4@SiO2 core-shell nanoparticles and the adsorption of CrO42‒, Ni2+, and Pb2+ ions on the polymer were investigated. It was found that the maximum adsorption conditions were pH = 6.5, ion concentration 250 mg/L, and adsorbent amount 0.075 g at room temperature. This paper is very interesting and meaningful for the development of recyclable adsorbents for toxic metals. I recommend that this paper is published after minor revisions according to a following comments.

Comments:

Page 4, Line 21

PEI (left) --> PEI (top)

NiFe2O4 (right) --> NiFe2O4 (bottom)

What does “Recoverd” mean in Figure 1 (bottom)?

Page 5, Lines 7-8

The compositional values of EDX analysis need to be discussed in detail. The composition formula of PEI is -[CH2CH2NH]-, so the weight ratio of C and N should be 24:14. However, the analysis value is C:N=36:39. Why is the nitrogen analysis value high?

Page 7, Lines 21-22

The authors mention XPS analysis, but is it possible to analyze the ions released during desorption?

Author Response

Dear Reviewer,

thank you very much for carefully reading our manuscript. We have revised it according your valuable comments.

Best wishes,

AXEL KLEIN

Reviewer 2 Report

Comments and Suggestions for Authors

Author Response

(The authors gave the same response as above.)

Reviewer 3 Report

Comments and Suggestions for Authors

- Title has words “grafted on nano”, for grafting process correct is “onto” my recommendation is change “grafted on nano” to “grafted onto nano”

- Introduction has many references for shorth sentence, here is one example (page 2, lines 8 and 9) “Frequently, the magnetic separation is based on hematite and magnetite embedded in core-shell nanoparticles [17,20,24–29]. 8 references for 15 words, try to include only important references, 2 more examples “[4,6,8–14].and [5,17,20–23]”

- Page 2, line 12, my recommendation is to change “grafted on” to “grafted onto” and the same recommendation for all manuscript

- Page 3 after line 42, Equation needs to include number

- Page 4, line 14 “ PEI polymer followed by reaction with core-shell” include evidences to confirm that authors obtained core-sell polymer

- Include information for Figure 1 if obtained crystalline materials or semi-crystalline materials and improve discussion for this Figure

- Figure 2, correct is “Transmittance” no “Transmission”, correct it, FT-IR Figure has Transmittance from 15 to 21.5, correct is from 0 to 100, normalize this spectrum with FT-IR tools or change “Transmittance (%)” to “Transmittance (a.u.)” and erase numbers. Note a.u. = arbitrary units

- Page 5, lines 13 and 14 “Thermogravimetric (TGA) and differential thermal analysis (DTA) showed a small weight loss of about 5% around 100 °C which is due to loss of adsorbed water”. TGA must dry for remove water or solvent before measure, so my recommendation is to remove this sentence

- Figure 4 discussion needs to include 10% weight loss and residual at 700 °C. Figure 4 needs to improve discussion, because in the present for looks very poor

- Page 6 line 7 “the pH war varied” maybe correct is “the pH was varied”

- In general, Figures need to improve at minimal 300 dpi, because Figures in the present form have very poor resolution, specially Figures 4-10

- Manuscript has good results but doesn’t have good discussion, improve discussion for all Figures

- Manuscript has 12 Figures but no mentioned in these Figures confirmation or grafted sample

- FT-IR and TGA-DTA have only one sample, why didn’t include more characterization samples 

Author Response

(The authors gave the same response as above.)

Round 2

Reviewer 2 Report

Comments and Suggestions for Authors

Dear authors,

I appreciate the time and dedication you took to respond to each of the comments made. I was pleased to be able to find correct answers to each of my suggestions. In my opinion, this new version of your manuscript can already be recommended for publication in Molecules.

Congratulations

Reviewer 3 Report

Comments and Suggestions for Authors

Manscript is accepting in the present form